# COVID-19 vaccine hesitancy among women planning for pregnancy, pregnant or breastfeeding mothers in Jordan: A cross-sectional study

**Rami Masa'deh**[1]*, **Aaliyah Momani**[1], **Ahmad Rayan**[2], **Shaher H. Hamaideh**[3], **Omayma M. Masadeh**[4], **Nabeel Al-yateem**[5]

1 School of Nursing, Applied Science Private University, Amman, Jordan, 2 School of Nursing, Zarqa University, Zarqa, Jordan, 3 Faculty of Nursing, Community and Mental Health Nursing Department, The Hashemite University, Zarqa, Jordan, 4 Relief International, Amman, Jordan, 5 Department of Nursing, College of Health Sciences, University of Sharjah, Sharjah, United Arab Emirates

* r_masadeh@asu.edu.jo

## Abstract

### Background

Women planning to become pregnant, who are pregnant, and who are breastfeeding are more hesitant to take COVID-19 vaccines compared to other women globally.

### Aim

This study investigates COVID-19 vaccine hesitancy among women, who are planning for pregnancy, currently pregnant, and breastfeeding women in Jordan

### Methods

An online cross-sectional study was conducted in the biggest three cities in Jordan, including 874 women.

### Results

Women who were planning for pregnancy, pregnant, or breastfeeding reported statistically significant lower levels of perception of the seriousness of COVID-19 (7.12 ± 0.72, 7.53 ± 1.80, 7.2439 ± 7296, respectively), significant lower levels of perceived benefits of the vaccine (8.92 ± 2.15, 8.73 ± 1.93, 9.09 ± 2.10, respectively), significant lower levels of motivation and causes of action (7.15 ± 1.71, 6.7524 ± 1.40, 7.27 ± 1.68, respectively), and significantly higher levels of COVID-19 vaccination hesitancy (31.32 ± 6.40, 30.11 ± 4.49, 30.27 ± 6.29, respectively) than other women.

Married women, those whoe were previously infected with COVID-19, and those who had chronic diseases reported statistically significant lower levels of perception of COVID-19 seriousness, perceived benefits of COVID-19 vaccine, motivation to take COVID-19 vaccine, and causes of action, and significantly higher levels of hesitancy to take COVID-19

**Data Availability Statement:** All relevant data are within the paper and its Supporting Information files.

**Funding:** This research project was funded by the Applied Science Private University, Amman, Jordan (https://www.asu.edu.jo/en/Pages/default.aspx) under grant number DRGS 2021-2022-9. Rami Masa'deh (RM) received the grant. The funder had no role in study design, data collection, and analysis, decision to publish, or preparation of the manuscript.

**Competing interests:** The Authors have declared no competing interests exist.

vaccine than unmarried women, those who have not been infected with COVID-19, and those who were medically healthy (p<0.001). There were statistically significant positive correlations between perception, perceived benefits, motivation, and cause of action with years of education; and statistically significant negative correlations between perception, perceived benefits, motivation, and cause of action with age (p<0.001).

## Conclusions

Women who were planning for pregnancy, pregnant, or breastfeeding in Jordan showed miderate scores in COVID-19 vaccine hesitancy despite the current international recommendations for its safety for women and their foetuses or neonates.

## 1. Introduction

The acute respiratory viral infection Coronavirus disease 2019 (COVID-19) rapidly spread from Wuhan, China, to become a global pandemic during early 2020 [1–4]. The severity of COVID-19 infection ranges from asymptomatic or very mild flu-like symptoms to fatal consequences. The World Health Organisation (WHO) reported that in very severe cases, patients may have multi-organ failure requiring hospitalisation [2]. The Centers for Disease Control and Prevention (CDC) reported that all individuals at any age can be infected with the virus and their infection can be severe; however, individuals with underlying medical conditions are at higher risk of having more severe consequences of COVID-19 [2,5].

The pandemic and the available vaccines have affected women's pregnancy planning behaviours [6]. Furthermore, One study in the UK showed that 267 out of 504 women reported that the pandemic affected their plans for pregnancy, and that 189 of them deliberately postponed their pregnancy due to the pandemic [7]. Moreover, the CDC reported that women who were pregnant were more likely to have a more severe form of COVID-19 compared to women who were not pregnant [5]. A study in the UK of 472 pregnant women diagnosed with COVID-19 and admitted to hospitals reported that 10% of them required respiratory support [8].

Although there is no evidence to suggest that COVID-19 can be transmitted through breast milk, breastfeeding women diagnosed with the virus reported that their main concern was transmitting it to their newborn [9]. It can be transmitted via aerosol during the breastfeeding process [10]. Consequently, the WHO and the CDC recommend that breastfeeding women diagnosed with COVID-19 should be encouraged to initiate or continue breastfeeding, as the benefits of breastfeeding far outweigh the potential risks of COVID-19 transmission [2,5].

Several COVID-19 vaccines (hereinafter "vaccines") were developed in an attempt to end the pandemic, four of which received emergency use authorisation approval to be used in Jordan, namely: Pfizer/ BioNTech, Sputnik V, Oxford/ AstraZeneca, and Sinopharm [2,11]. Historically, vaccines are considered major success stories that saved and continue to save millions of lives around the globe. However, some people are still hesitant to take vaccines [12]. Vaccine hesitancy was defined as "delay in acceptance or refusal of vaccination despite availability of vaccination services" [13].

COVID-19 vaccination is recommended for women who are planning for pregnancy, pregnant, and breastfeeding women [5]. However, vaccine acceptability is lower among pregnant women or planning to get pregnant compared to non-pregnant women [14]. Several studies in Arabic and non-Arabic countries showed that pregnant women and planning to be pregnant were more hesitant to take vaccine compared to women who were not pregnant or planning to

be pregnant [15,16]. One meta-analysis estimated vaccine hesitancy of 48.4% among pregnant and breastfeeding women [17]. This hesitancy was explained by two main reasons. First, there are popular concerns regarding vaccine safety, because it was developed in a short period of time and received accelerated emergency approval for use [16,18]. This created concerns about potential harms for the foetus among pregnant women, or infertility for women who were planning to get pregnant [15]. Second, a perceived lack of knowledge of the "novel" COVID-19 virus among the public (and to some extent an actual lack of knowledge of its long-term impacts) causes many women to fear its effects on pregnancy and breastfeeding, with uncertainty about how vaccines can minimise the severe effects of the disease on pregnant and breastfeeding women [16,19]. Several factors can affect vaccine hesitancy among pregnant and breastfeeding women, this could include: religion; interpersonal norms; risk perception; role of healthcare providers; availability and accesability of the vaccine [20].

Vaccine hesitancy among these women can have negative consequences on them and their foetus or neotes. Therefore, the aim of this study was to investigate COVID-19 vaccine hesitancy among women who are planning for pregnancy, currently pregnant, and breastfeeding women in Jordan

## 2. Methods

### 2.1. Study design

This study implements an online cross-sectional study.

### 2.2. Setting

The study was conducted in the largest three cities of Jordan (Amman, Irbid, and Zarqa), which include more than three-quarters of the national population [21]. Data were collected in the period between 1st December 2021 and 15th January 2022. Twelve maternal clinics representing all health sectors (governmental, private, and educational) were involved.

### 2.3. Population, sampling strategy, and sample size

The target population was all adult women in Jordan. A non-probability convenience sampling technique was used to recruit women. The inclusion criteria were women who are above 18 years old; and able to read, write and understand Arabic.

The sample size was calculated using G power software [22]. Taking into consideration the tests that are used in this study, with a power of 0.95, medium effect size of 0.25, and 0.05 level of significance, a total of 324 women were required to obtain statistically significant results. However, the study included two groups of women, the first group included women who were planning for pregnancy (trying to get pregnant or thinking of it); currently pregnant(with positive signs of pregnancy); or currently breastfeeding (breastfeeding their children exclusively or not regardless the age of their child). The second group includedwomen who did not fall into the first group categories (planning to be pregnant, pregnant or breastfeeding). The main purpose of the study was to examine perception of the seriousness of COVID-19, vaccine hesitancy, perceived benefits of the vaccine, motivation and cause of action in women who are planning for pregnancy, pregnant, and breastfeeding, and to compare them with other women. Therefore, it was important to increase the sample size to increase the likelihood of representing those women. Additionally, due to the lack of research on this issue in Jordan, a national study was needed with a large sample size. Consequently, it was deemed important to increase the sample size, and 847 women were ultimately included. In this study, 877 accessed the questionaires and 847 fully competed the study.

## 2.4. Measurements of variables

**2.4.1. Demographic characteristics sheet.** The characteristics sheet concerns twelve question about general data such as the women's age, city of residence, educational level, marital status, and financial status, and health-related information concerning whether they are planning for pregnancy, pregnant, or breastfeeding women, history of chronic diseases, COVID-19 vaccination history, previous infection with COVID-19.

**2.4.2. The Arabic version of perceptions, hesitancy, perceived benefits, motivation and cause of action about COVID-19 vaccination questionnaire.** This questionnaire has been previously used in studies around the globe [23,24], and has been reported to be a valid and a reliable questionnaire to be used in Arabic communities [16,24]. The questionnaire covers four dimensions: perceptions about the seriousness of COVID-19 (three items), hesitancy about receiving the vaccine (nine items), perceived benefits of receiving the vaccine (four items),and motivations and causes of action for receiving the vaccine (three items).

Each item is scored using a five-point Likert scale, ranging from 1 (strongly disagree) to 5 (strongly agree). The possible scores for perception ranged from 3 to 15; for reasons for hesitancy ranged from 9 to 45; perceived benefits ranged from 4 to 20; and motivation and causes of action ranged from 3 to 15. Higher scores in each domain indicate higher perception about the seriousness of the COVID-19 situation; hesitancy to receive the vaccine; perception about the perceived benefits of the vaccine; and motivation or likelihood to take action to receive the vaccine (respectively) [16,24].

In this study, perceptions of COVID-19 seriousness, vaccine hesitancy, perceived benefits, and motivation and causes of action to take the vaccine were all considered as dependent variables (outcomes) whereas other variables were grouping variables. In this study, the Cronbach's alpha reliability for this tool was examined and found to be good (0.87).

## 2.5. Ethical considerations

Ethical approval was obtained from the ethical committee of the Applied Scince Private University (IRB number: Faculty 2021-2022-1-9) and access was granted from all participating organizations. Before collecting data, participants were informed that participation was voluntary and they had the right to withdrawal at any stage without giving a reason. Upon conducting the study, participants were informed about the anonymity and confidentiality measures taken, and were assured that all data would be used only for research purposes; furthermore, no one other than the researchers would have access to their data. Each participant was assigned a digital code number to protect their identity. Moreover, there was an written/online consent form that included the title and purpose of the study that women had to agree before being involved in the study. By agreeing on the digital consent form, women declared that everything was clear to them, and that they willingly agreed to participate in the study. The online-completed questionnaires were stored in a secured desktop accessible only to the principal researcher.

## 2.6. Data collection procedure

After gaining the required ethical approvals, posters were announced at several maternity clinics that includes a link and a barcode where invited women can click in to participate. They were then presented with an invitation letter explaining the purpose of the study and encouraging them to voluntarily participate, by explaining the aim and the national importance of the study (no incentive was provided for participation). This link allowed only those who met the inclusion criteria to fill in the questionnaires. An IT professional designed the barcode in a way that allowed participants to answer the questionnaire only once from the same device. To ensure the qulity of data provided and the accuracy of the answers to avoid missing data, the link provided

to the participants allows submission only when all answers are completed. At the end of the questionnaire participants submitted their answers, which were automatically sent to the principal researcher. Completing the questionnires took around 10 minutes from each participant and all participants answered the questionnires themselves without needing assistant.

The material was piloted for 10% of the sample size. All questions were clear and there was no modification to the process/instruments. Therefore, those were included in the final analysis.

## 2.7. Statistical analysis

Data were analysed using SPSS version 27 [25]. Categorical variables were reported as frequencies and percentages. Continuous data were reported as mean ± standard deviation (SD).

**Table 1. Characteristics of participating women.**

| Variable | Mean ± SD or frequency (%) N=874 |
|---|---|
| **Age** | **36.63 ± 10.18** |
| **Marital status:** | |
| Single | 348 (39.8%) |
| Married | 492 (56.3%) |
| Divorced | 24 (2.7%) |
| Widowed | 10 (1.1%) |
| **Are you:** | |
| Planning for pregnancy | 136 (15.6%) |
| Pregnant women | 210 (24.0%) |
| Breastfeeding women | 82 (9.4%) |
| Others | 446 (51.0%) |
| **COVID-19 vaccination:** | |
| Not taken | 234 (26.8%) |
| One dose | 178 (20.4%) |
| Two doses | 290 (33.2%) |
| Three doses | 172 (19.7%) |
| **Previous COVID-19 infection:** | |
| Yes | 490 (56.1%) |
| No | 384 (43.9%) |
| **Chronic diseases:** | |
| No medical history | 696 (79.6%) |
| Hypertension | 52 (5.9%) |
| Diabetes mellitus | 56 (6.4%) |
| Immune system disease | 28 (3.2%) |
| Others | 42 (4.8) |
| **Employment:** | |
| Employed | 510 (58.4%) |
| Unemployed | 364 (41.6%) |
| **Residency:** | |
| Amman | 470 (53.8%) |
| Zarqa | 247 (28.3%) |
| Irbid | 157 (18%) |
| **Level of education:** | |
| Primary education | 26 (3%) |
| Secondary education | 48 (5.5%) |
| Bachelor | 757 (86.6%) |
| High education | 43 (4.9%) |
| **Financial status:** | |
| Comfortable | 210 (24.03%) |
| Varies | 290 (33.18%) |
| Tight | 374 (42.79%) |
| **Years of education** | ± 2.71 |

Independent samples t-test and one-way analysis of variance (ANOVA) were used to compare the mean scores of the perception, hesitancy, benefits and motivation and cause of action of vaccine decisions between different variables. Moreover, Pearson r correlation test was used to find relationships between variables. However, before running all previous inferential statistics, the assumption required for each test were tested and guaranteed. Normall distribution was checked for the main variables and all of them were normally distributed. A p-value of <0.05 was applied to represent the statistical significance of the results, and the level of significance was predetermined as 5%.

## 3. Results

### 3.1. Participant characteristics

A total of 874 women completed the questionnaire and were involved in the final analysis in this study, Table 1. Their ages ranged from 19 to 52 years with a mean of 36.63 ± 10.18, and the majority of them reported having no chronic diseases. More than half of the women were married (56%) and employed (58.4%). Almost half of them were either for pregnancy, pregnant, or breastfeeding women. Approximately a quarter of them did not get a vaccine, while the rest had received either one dose, two doses, or three doses. In terms of previous COVID-19 infection, 56.1% had previously been infected with COVID-19.

### 3.2. Dependent variables: Perceptions of COVID-19 seriousness, vaccine hesitancy, perceived benefits, and motivation and causes of action to take the vaccine

As presented in Table 2, women in this study showed moderate scores in terms of their perception, hesitancy, perceived benefits, and motivation and causes of action of being vaccinated. The mean perception score for all participants was 8.78 (SD: 2.7) out of 15. The mean hesitancy score for all women was 26.59 (SD: 7.86) out of 45. The mean perceived benefits score for all women was 11.24 (SD: 3.56) out of 20. The mean causes of action score for all participants was 8.94 (SD: 2.87) out of 15.

### 3.3. Differences in dependent variables by grouping variables

One-way analysis of variance (ANOVA) and post hoc analysis were conducted to examine differences in perceptions of the seriousness of COVID-19, vaccine hesitancy, perceived benefits of the vaccine, and motivation and causes of action about taking the vaccine according to pregnancy status, residency and financial status of the women. As presented in Table 3, results showed that women planning for pregnancy, pregnant, and breastfeeding reported statistically significant lower levels of perception of the seriousness of COVID-19 (7.12 ± 0.72, 7.53 ± 1.80, 7.2439 ± 7296, respectively), lower levels of perceived benefits of the vaccine (8.92 ± 2.15, 8.73 ± 1.93, 9.09 ± 2.10, respectively), lower levels of motivation and causes of action (7.15 ± 1.71, 6.7524 ± 1.40, 7.27 ± 1.68, respectively), and significantly higher levels of COVID-

**Table 2. Perception of the disease, vaccine hesitancy, benfits and motivation.**

| Variable | Mean ± SD |
|---|---|
| **Perception of seriousness of COVID-19** | 8.78 ± 2.70 |
| **Vaccine hesitancy** | 26.59 ± 7.86 |
| **Perceived benefits of vaccine** | 11.24 ± 3.65 |
| **Motivation and cause of action of taking vaccine** | 8.94 ± 2.87 |

**Table 3. Differences in perceptions of the seriousness of COVID-19, vaccine hesitancy, perceived benefits of the vaccine, and motivation and causes of action about taking the vaccine according to pregnancy status of the women.**

| Dependent variable Mean ± SD | Grouping variable: pregnancy status Mean ± SD | | Mean difference | P value | F |
|---|---|---|---|---|---|
| Perception 8.78 ± 2.70 | pregnant women 7.53 ± 1.80 | planning to be pregnant 7.12 ± 0.72 | 0.42 | .103 | 108.73 |
| | | breastfeeding women 7.24 ± 0.73 | 0.29 | .336 | |
| | | Others 10.6 ± 2.94 | -2.62 | .000 | |
| | planning for pregnancy 7.12 ± 0.72 | breastfeeding women 7.24 ± 0.73 | -0.01 | .696 | |
| | | Others 10.6 ± 2.94 | -3.04 | .000 | |
| | breastfeeding women 7.2439 ± 7296 | Others 10.1570 ± 2.9417 | -2.91 | .000 | |
| Hesitancy 26.5881 ± 7.85957 | pregnant women 30.11 ± 4.49 | planning to be pregnant 31.32 ± 6.40 | -1.21 | .109 | 93.49 |
| | | breastfeeding women 30.27 ± 6.29 | -.154 | .863 | |
| | | Others 22.81 ± 7.91 | 7.31 | .000 | |
| | planning for pregnancy 31.32 ± 6.40 | breastfeeding women 30.27 ± 6.29 | 1.06 | .271 | |
| | | Others 22.81 ± 7.91 | 8.52 | .000 | |
| | breastfeeding women 30.27 ± 6.29 | Others 22.81 ± 7.91 | 7.46 | .000 | |
| Benefits 11.2414 ± 3.55823 | pregnant women 8.73 ± 1.93 | planning to be pregnant 8.92 ± 2.15 | -0.19 | .522 | 221.33 |
| | | breastfeeding women 9.09 ± 2.10 | -0.36 | .304 | |
| | | Others 13.53 ± 3.18 | -4.81 | .000 | |
| | planning for pregnancy 8.92 ± 2.15 | breastfeeding women 9.09 ± 2.10 | -0.17 | .651 | |
| | | Others 13.53 ± 3.18 | -4.62 | .000 | |
| | breastfeeding women 9.09 ± 2.10 | Others 13.53 ± 3.18 | -4.45 | .000 | |
| Motivation and cause of action 8.9405 ± 2.86722 | pregnant women 6.7524 ± 1.40 | planning to be pregnant 7.15 ± 1.71 | -0.39 | .092 | 240.86 |
| | | breastfeeding women 7.27 ± 1.68 | -0.52 | .062 | |
| | | Others 10.83 ± 2.55 | -4.07 | .000 | |
| | planning for pregnancy 7.15 ± 1.71 | breastfeeding women 7.27 ± 1.68 | -0.12 | 683 | |
| | | Others 10.83 ± 2.55 | -3.68 | .000 | |
| | breastfeeding women 7.27 ± 1.68 | Others 10.83 ± 2.55 | -3.56 | .000 | |

19 vaccination hesitancy (31.32 ± 6.40, 30.11 ± 4.49, 30.27 ± 6.29, respectively) than other women (perception 10.6 ± 2., perceived benefits 13.53 ± 3.18, motivation and causes of action 10.83 ± 2.55, and hesitancy 22.81 ± 7.91, respectively; p<0.001). However, the results showed no significant difference in perception, hesitancy, perceived benefits, and motivation and causes of action of women according to their residency and financial status.

Independent samples t-test was used to examine the difference in the perception of COVID-19 seriousness, vaccine hesitancy, perceived benefits, and motivation and causes of action of taking the vaccine according to previous infection with COVID-19, history of chronic diseases, and marital status. Table 4 shows that women who were previously infected with COVID-19 reported significantly lower levels of perception (8.40 ± 2.56), perceived benefits (10.18 ± 3.30), and motivation and causes of action (8.05 ± 2.65); and significantly higher levels of hesitancy (29.42 ± 6.82) than those who have not been infected with COVID-19 (perception 9.27 ± 2.81, perceived benefits 12.59 ± 3.42, motivation and cause of action 10.07 ± 2.74, hesitancy 22.97 ± 7.61).

Secondly, concerning history of chronic diseases, women were divided into two groups: those who had chronic disease(s); and those who were medically healthy. The results in Table 4 show that the former reported statistically significant lower levels of perception (8.03 ± 2.21), perceived benefits (7.38 ± 1.54), and motivation and causes of action (9.42 ± 1.95) than those who were medically healthy (perception 8.97 ± 2.79, benefits

**Table 4. Difference in the perception of COVID-19 seriousness, vaccine hesitancy, perceived benefits, and motivation and causes of action of taking the vaccine according to previous infection with COVID-19, history of chronic diseases and marital status.**

| Variable | Grouping variable | | Mean ± SD | P value | T |
|---|---|---|---|---|---|
| Perception | Previous infection: | Yes (490) | 8.40 ± 2.56 | .000 | -4.754 |
| | | No (384) | 9.27 ± 2.81 | | |
| Hesitancy | Previous infection: | Yes (490) | 29.42 ± 6.82 | .000 | 3.192 |
| | | No (384) | 22.97 ± 7.61 | | |
| Benefits | Previous infection: | Yes (490) | 10.18 ± 3.30 | .000 | -10.534 |
| | | No (384) | 12.59 ± 3.42 | | |
| Motivation | Previous infection: | Yes (490) | 8.05 ± 2.65 | .000 | 11.027 |
| | | No (384) | 10.07 ± 2.74 | | |
| Perception | Chronic diseases: | Yes (178) | 8.03 ± 2.21 | .000 | -4.17 |
| | | No (696) | 8.97 ± 2.79 | | |
| Hesitancy | Chronic diseases: | Yes (178) | 28.60 ± 6.67 | .000 | 3.84 |
| | | No (696) | 26.07 ± 8.06 | | |
| Benefits | Chronic diseases: | Yes (178) | 7.38 ± 1.54 | .000 | -8.45 |
| | | No (696) | 9.30 ± 2.30 | | |
| Motivation | Chronic diseases: | Yes (178) | 9.42 ± 1.95 | .000 | -7.90 |
| | | No (696) | 11.71 ± 3. 72 | | |
| Perception | Marital status: | Married (492) | 7.19 ± 1.35 | .000 | 26.58 |
| | | Unmarried (382) | 10.83 ± 2.62 | | |
| Hesitancy | Marital status: | Married (492) | 30.11 ± 5.41 | .000 | -17.51 |
| | | Unmarried (382) | 22.04 ± 8.19 | | |
| Benefits | Marital status: | Married (492) | 9.17 ± 2.52 | .000 | 25.98 |
| | | Unmarried (382) | 13.91 ± 2.86 | | |
| Motivation | Marital status: | Married (492) | 7.24 ± 1.99 | .000 | 26.79 |
| | | Unmarried (382) | 11.13 ± 2.29 | | |

9.30 ± 2.30, motivation and cause of action 11.71 ± 3. 72) (p<0.001). Women who had chronic diseases showed significantly higher levels of hesitancy (28.60 ± 6.67) than those who were medically healthy (26.07 ± 8.06) (p<0.001).

Thirdly, the differences in perceptions of the seriousness of COVID-19, hesitancy to take the vaccine, perceived benefits of the vaccine, and motivation and causes of action according to marital status were examined. The vast majority of women were either single or married; very few of them reported being either widowed or divorced. Therefore, marital status was recoded to "married" versus "unmarried". The results showed that married women had significantly lower perceptions of seriousness (7.19 ± 1.35), benefits of vaccine (9.17 ± 2.52), and motivation and cause of action (7.24 ± 1.99) than their unmarried counterparts (perception 10.83 ± 2.62, benefits 13.91 ± 2.86, motivation and cause of action 11.13 ± 2.29) (p<0.001). Married women reported significantly higher hesitancy (30.11 ± 5.41) than unmarried women (22.04 ± 8.19) (p<0.001) (Table 4).

## 3.4. Relationships between dependent and grouping variables

The relationship between perception of seriousness of COVID-19, vaccine hesitancy, perceived vaccine benefits, motivation and causes of action, age of women and years of education were examined using Pearson r correlation coefficient. The results (Table 5) indicate statistically significant positive correlations between perception, perceived benefits, and motivation and cause of action with years of education (r=.23, r=.11, r=.11, respectively: p<0.001). On the

other hand, statistically significant negative correlations were found between perception, perceived benefits, and motivation and cause of action with age (r=-.58, r=-.25, r=-.262, respectively: p<0.001). In terms of hesitancy, results showed a significant positive relationship between hesitancy and age r=.29, (p<0.001), and a significant negative relationship between hesitancy and years of education r=-.14, (p<0.001).

## 4. Discussion

The aim of this study was to investigates COVID-19 vaccine hesitancy among women, who are planning for pregnancy, currently pregnant, and breastfeeding women in Jordan. The study included women who are able to read and write Arabic and also have access to internet by their device. In Jordan, the illiteracy rate for this age group is lower than 1% and due to the fact that almost quarter of the participating women did not take the vaccine so it was deemed important to minimize the risk of being with them during the data collection especially during the pandamic of COVID19, so the research team suggested to include only those group of women. The study showed that women who were planning for pregnancy, who were pregnant, or who were breastfeeding reported statistically significant lower levels of perception of the seriousness of COVID-19, significant lower levels of perceived benefits of the vaccine, significant lower levels of motivation and causes of action, and significantly higher levels of COVID-19 vaccination hesitancy than other women. Furthermore, married women, those whoe were previously infected with COVID-19, and those who had chronic diseases reported statistically significant lower levels of perception, perceived benefits, motivation, and causes of action, and significantly higher levels of hesitancy than unmarried women, those who have not been infected with COVID-19, and those who were medically healthy. There were statistically significant positive correlations between perception, perceived benefits, motivation, and cause of action with years of education; and statistically significant negative correlations between perception, perceived benefits, motivation, and cause of action with age. There was a significant positive relationship between hesitancy and age, and a significant negative relationship between hesitancy and years of education.

The results of the current study showed that Jordanian women scored moderate levels of perception of the seriousness of COVID-19, COVID-19 vaccine hesitancy, perceived benefits, and motivation and causes of action of being vaccinated. Vaccine hesitancy was reported in global studies in the US and Europe to range between 30-77% [26–29]. Moreover, a study conducted in Egypt showed that 196 out of 385 participants were undecided about vaccination, 108 out of 385 refused it, and only 81 accepted to take it [30]. A similar situation was reported in Jordan, where only a quarter of 3000 participants accepted vaccination [31]. It seems that vaccine hesitancy in relation to the COVID-19 pandemic is a common global phenomenon, and such hesitancy might be particularly high in specific situations [32]. This might be explained that such group of women are not well expsed to the available data regarding the COVID-19 and the available vaccines. Therefore, there is a crucial role on the media to transefer these information scientifically.

The current study showed that women who were married, planning forpregnancy, pregnant or breastfeeding had higher COVID-19 vaccination hesitancy compared to other women. This result is in line with several studies around the globe which showed that women who were married, planning for pregnantcy, pregnant, and breastfeedin, were more vaccine hesitant than other women [26,33,34]. Even though the Royal College of Obstetricians and Gynaecologists, the WHO, and the CDC strongly recommend vaccines and booster doses for women planning for pregnancy, pregnant, and breastfeeding, the acceptance of the vaccine by those women remains relatively low [2,5,35]. Investigating this further, the results of the current

**Table 5. Relationship between perception of seriousness of COVID-19, vaccine hesitancy, perceived vaccine benefits, motivation and causes of action, age of women and years of education.**

| Variable | | Perception | Hesitancy | Benefits | Motivation | Years of education | Age |
|---|---|---|---|---|---|---|---|
| Perception | Pearson r correlation | | -.21 | .35 | .36 | .23 | -.58 |
| | P value | | .000 | .000 | .000 | .000 | .000 |
| Hesitancy | Pearson r correlation | -.21 | 1 | -.26 | -.27 | -.14 | .29 |
| | P value | .000 | | .000 | .000 | .000 | .000 |
| Benefits | Pearson r correlation | .35 | -.26 | 1 | .98 | .11 | -.25 |
| | P value | .000 | .000 | | .000 | .001 | .000 |
| Motivation | Pearson r correlation | .36 | -.27 | .98 | 1 | .11 | -.262 |
| | P value | .000 | .000 | .000 | | .002 | .000 |
| Years of education | Pearson r correlation | .23 | -.16 | .11 | .11 | 1 | -.55 |
| | P value | .000 | .000 | .001 | .002 | | .000 |
| Age | Pearson r correlation | -.58 | .29 | -.25 | -.26 | -.55 | 1 |
| | P value | .000 | .000 | .000 | .000 | .000 | |

study show that perceptions of the seriousness of COVID-19, perceived benefits, and motivation and causes of action of being vaccinated are all positively correlated, but all of them are negatively correlated with hesitancy. These findings suggest that increasing awareness about the seriousness of COVID-19, vaccine benefits, and motivation to take the vaccine will decrease vaccine hesitancy.

The results of the current study also showed that women with more years of education had higher levels of perception of the seriousness of COVID-19, perceived benefits, motivation and cause of action, and lower level hesitancy to take the vaccine. This is similar to a study that was conducted in Canada, which reported that women with less than a bachelor's degree were more hesitant to take the vaccine compared to women with higher levels of education [36]. Similarly, a systematic review of nine studies from 24 countries including Europe and United States showed that vaccine acceptance among pregnant women and breastfeeding women was higher among those with higher levels of education [37].

In the current study, a negative correlation was found between age and perception of the seriousness of COVID-19, perceived benefits, motivation and cause of action, whereas a positive correlation was found between hesitancy and age. On one hand, age was not a factor in terms of vaccine hesitancy [36]. On the other hand, one study that was conducted in Australia showed that hesitancy to take vaccine was lower among older individuals when compared to younger ones [38]. In the US and Europe, older age was found to be a co-existing factor with vaccine hesitancy among pregnant women and breastfeeding women [37]. These inconsistencies in the literature may suggest that further research is needed to understand how age influence vaccine hesitancy.

Women who participated in the current study who had previously been infected with COVID-19 had lower levels of perception of the seriousness of COVID-19, perceived benefits, motivation and causes of action and higher levels of hesitancy when compared to women who did not have a previous infection with COVID-19. This is consistent with a large-scale study that examined over half a million adults in the US which showed that being infected with COVID-19 halves the odds of receiving the vaccine [39]. It seems that there is a belief that being infected with COVID-19 gives individuals immunity against the virus, and this increases hesitancy to take the vaccine.

The results showed that women diagnosed with chronic disease(s) had lower levels of perception of the seriousness of COVID-19, perceived benefits, motivation and causes of action,

and higher levels of hesitancy than those who did not have such disease(s). This differs from a large-scale Australian study of 1420 individuals which showed that individuals with chronic diseases reported higher acceptance to take the vaccine [38]. Despite that fact that the seriousness of the disease and the risk of hospitalization or even death among those with chronic diseases is higher compared to those without chronic diseases, as reported by several global organizations such as WHO and CDC, women with chronic diseases in this study reported high levels of hesitancy. This might be explained as women planning for pregnancy, pregnant, or breastfeeding in this study being either unaware of these global recommendations, or requiring more reassurance to facilitate informed pregnancy vaccine decisions [40–42].

This study has several implications, it is important that policy-makers focus on women who are planning for pregnancy, currently pregnant, or currently breastfeeding in Jordan in terms of providing appropriate evidence-based health education on the seriousness of the COVID-19 disease and the importance of COVID-19 vaccines. Furthermore, it is important for health-care providers to educate women on the importance of COVID-19 vaccines.

## 5. Stengths and limitations

The major strength of this study is it novelty to the context of Jordan, and a relatively large sample size. The major limitation of this study is the nature of quastionnare distribution, which was online due to pandemic restrictions. Furthermore, the study was limited by including women who were literate and who have smartphones with access to the internet. This should be explored more in future research.

## 6. Conclusions

In conclusion, the results of the current study showed that women, who were planning for pregnancy, pregnant, or were breastfeeding in Jordan were hesitant to take COVID-19 vaccines despite international recommendations. Therefore, it is important that policy-makers and media focus on this group of women in Jordan in terms of providing appropriate health education on the importance of COVID-19 vaccines to decrease vaccine hesitancy among these women.

## Supporting information

**S1 Table. Characteristics of participating women.**
(DOCX)

**S2 Table. Perception of the disease, vaccine hesitancy, benfits and motivation.**
(DOCX)

**S3 Table. Differences in perceptions of the seriousness of COVID-19, vaccine hesitancy, perceived benefits of the vaccine, and motivation and causes of action about taking the vaccine according to pregnancy status of the women.**
(DOCX)

**S4 Table. Difference in the perception of COVID-19 seriousness, vaccine hesitancy, perceived benefits, and motivation and causes of action of taking the vaccine according to previous infection with COVID-19, history of chronic diseases and marital status.**
(DOCX)

**S1 File.**
(SAV)

## Author Contributions

**Conceptualization:** Rami Masa'deh.

**Data curation:** Rami Masa'deh, Aaliyah Momani, Shaher H. Hamaideh, Omayma M. Masadeh.

**Formal analysis:** Rami Masa'deh, Ahmad Rayan.

**Methodology:** Rami Masa'deh, Aaliyah Momani, Ahmad Rayan, Shaher H. Hamaideh, Omayma M. Masadeh, Nabeel Al-yateem.

**Project administration:** Rami Masa'deh.

**Writing – original draft:** Rami Masa'deh, Aaliyah Momani, Omayma M. Masadeh.

**Writing – review & editing:** Rami Masa'deh, Aaliyah Momani, Nabeel Al-yateem.

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
