## [Decision Letter · Decision Letter 0]

5 Aug 2022

PONE-D-22-08549COVID-19 vaccine hesitancy among Jordanian women who are planning to be pregnant, are pregnant or are breastfeedingPLOS ONE

Dear Dr. Masa'Deh,

Thank you for submitting your manuscript to PLOS ONE. After careful consideration, we feel that it has merit but does not fully meet PLOS ONE’s publication criteria as it currently stands. Therefore, we invite you to submit a revised version of the manuscript that addresses the points raised during the review process. Please ensure that you thoroughly address the reviewers comments and concerns and ensure the manuscript is free of any editorial or grammatical errors.

We look forward to receiving your revised manuscript.

Kind regards,

Mohamad Alameddine, MPH, Ph.D.

Academic Editor

PLOS ONE

https://journals.plos.org/plosone/s/file?id=ba62/PLOSOne_formatting_sample_title_authors_affiliations.pdf".

“Acknowledgment: The authors are grateful to the Applied Science Private University, Amman, Jordan, for full financial support granted to this research project

Funding statement: This research project was funded by the Applied Science Private University, Amman, Jordan.”

4. Thank you for stating the following in the Funding Section of your manuscript:

“: This research project was funded by the Applied Science Private University, Amman, Jordan.”

“Acknowledgment: The authors are grateful to the Applied Science Private University, Amman, Jordan, for full financial support granted to this research project

Funding statement: This research project was funded by the Applied Science Private University, Amman, Jordan.”

Additional Editor Comments:

Make sure you thoroughly address the reviewers comments and concerns and ensure the manuscript is free of any editorial or grammatical errorsز

Reviewers' comments:

Reviewer's Responses to Questions

**Comments to the Author**

1. Is the manuscript technically sound, and do the data support the conclusions?

Reviewer #1: Yes

Reviewer #2: Partly

Reviewer #3: Partly

Reviewer #4: Yes

2. Has the statistical analysis been performed appropriately and rigorously? 

Reviewer #1: Yes

Reviewer #2: No

Reviewer #3: Yes

Reviewer #4: I Don't Know

3. Have the authors made all data underlying the findings in their manuscript fully available?

Reviewer #1: Yes

Reviewer #2: No

Reviewer #3: No

Reviewer #4: No

4. Is the manuscript presented in an intelligible fashion and written in standard English?

Reviewer #1: Yes

Reviewer #2: Yes

Reviewer #3: No

Reviewer #4: Yes

5. Review Comments to the Author

Reviewer #1: It is an interesting and timley study and informs advocacy policy to a good extent. Unfortunately and due to the nature of the design and methodology, it cant inform what in the pregnant category made them hesitant (fetal risks, mutation, maternal side effects, etc). This could be something to consider in future studies to taylor promotion messages

Reviewer #2: This study assessed COVID-19 vaccine hesitancy among Jordanian women who are planning to be pregnant, are pregnant or are breastfeeding.

I did not see any scientific merit of the study. Indeed the analysis was simple but not strong

The researcher made assumptions that women did not receive information. You have to consider that this is a new pandemic and little evidence is available. So hesitancy is an expected results

Reviewer #3: Although the idea behind conducting the hesitancy is appreciable, but just limiting to if vaccine hesitancy is there won't be a sufficient to add value to already available scientific works.

Why the hesitancy is there should be looked into.

Also the demographic variables should be most elaborated like year of education may not be same every where globally. Like professional courses/schooling/graduation etc

Chronic diseases could be looked into system wise involvements: Hypertension may not be a disease all alone but a risk factor for many chronic diseases affecting central nervous system, Renal and Cardiovascular system.

The more stringent description and definition of variables will increase granularity of the result and increase value of the findings which can be translated to policy development to devise solution of such societal perceptions towards any new therapeutics regimen introduced in coming days.

Previous covid 19 infection: Is it laboratory confirmed or not should be looked into. Also are there any vaccine breakthrough covid 19 infections/ reinfection after intial dose of covid 19 vaccine will be more interesting to look into. If these breaktrhough infection also played role in changing the perception and attitude.

Looking into family history of Vaccination/vaccine hesitancy, breakthrough infections are also important factors while studying hesitancy.

Digital health and role of Media over vaccine acceptance need to be studied correlating the level of education to see if any interesting aspect may come out. You may refer to one work published in Plos one

https://pubmed.ncbi.nlm.nih.gov/35180249/

The english should be proof read by native English speaking person to increase the flow and connection

Reviewer #4: The article produces important evidence regarding uptake of COVID-19 among women planning to become pregnant, who are pregnant, and who are breastfeeding which is key in formulating polices and recommendations to control the pandemic. However, the article cannot published in its current format due to a number of issues.

Introduction

1. The introduction seems to be long (2 pages). The authors should consider making short and concise ( introduce the topic, what is already known, what is unknown/justification for the study and finally aim/objective)

Methods

Study design

2. The authors mentions that it was “correlational” study design. Why do they refer the study that way? In correlational studies, the study unit is not people but geographical.

3. Some information is missing for sample size calculation. Consult a Statistician for the sample size calculation for cross-sectional studies with a comparison group. Some assumptions especially the two proportions (p1 and p2 ) and there references are missing.

4. “…..women who are planning to be pregnant, pregnant women, and breastfeeding women, and to compare them with other women”. Identifying pregnant and breastfeeding women. What was the distinction between women planning to be pregnant and other women? How were women planning to be pregnancy identified? There is a danger of misclassification.

5. Please include the name of ethical institution and the IRB number

6. How was the barcode/ link for the survey distributed to the participants? Was the scanning done via a mobile phone, bar code scanner etc? Please specify these details sequentially. Was measures were put in place that each participate answers the questionnaire once. What measures were also put to make sure those outside the study area did not participate? With social media link can be easily shared.

Results

7. The sample size calculation assumptions were to compare the outcomes of women who are planning to be pregnant, pregnant women, and breastfeeding women with the “other women”. The “other women” are not coming out in the results showing the comparison

8. The calculated sample size was 847 but you interviewed more than double 1, 874. Any implications and if yes these should be discussed.

9. In Table 1, were age and years of education normally distributed? The mean and SD is reported for normally distributed data, otherwise report the median.

10. Results section 3.2, “In this study, perceptions of COVID-19 seriousness, vaccine hesitancy, perceived benefits, and motivation and causes of action to take the vaccine were all considered as dependent variables whereas other variables were grouping variables”. This should be moved to the methods sections on variables and expanded further. Consider using the term outcome(s) and independent (exposure, explanatory) variables which easier to follow.

Discussion

11. The first paragraph of the discussion should summarise the main findings of the study and later paragraphs should discuss finding by finding.

12. Include strengths of the study

What is listed as limitations are not limitations. Usually limitations should include methodological issues which could have affected the validity of the results. Please revise

Others

13. The authors should maintain the sequence “Women planning to become pregnant, who are pregnant, and who are breastfeeding” throughout the paper. In some instances it not followed.

6. PLOS authors have the option to publish the peer review history of their article (what does this mean?). If published, this will include your full peer review and any attached files.

Reviewer #1: No

Reviewer #2: No

Reviewer #3: **Yes: **Mainak Bardhan

Reviewer #4: **Yes: **Richard Makurumidze

---

## [Author Response · Author response to Decision Letter 0]

21 Sep 2022

Response was submitted in a separate file

---

## [Decision Letter · Decision Letter 1]

5 Jan 2023

PONE-D-22-08549R1COVID-19 vaccine hesitancy among Jordanian women who are planning to be pregnant, are pregnant or are breastfeedingPLOS ONE

Dear Dr. Masa'Deh,

Thank you for submitting your manuscript to PLOS ONE. After careful consideration, we feel that it has merit but does not fully meet PLOS ONE’s publication criteria as it currently stands. Therefore, we invite you to submit a revised version of the manuscript that addresses the points raised during the review process.

We look forward to receiving your revised manuscript.

Kind regards,

Ghada Abdrabo Abdellatif Elshaarawy, M.D

Academic Editor

PLOS ONE

Journal Requirements:

Additional Editor Comments:

Additional Editor Comments:

1) Title:

The title is not scientifically written. It needs to be revised.

2) ABSTRACT:

It is too long; it needs to be shortened.

3) INTRODUCTION:

The text of the introduction is too long with redundancy and diffusion.

Newest Global/ Regional/ Jordan prevalence of COVID-19 vaccine hesitancy among pregnant and breastfed Jordanian women should be stated.

Risk factors associated with COVID-19 vaccine hesitancy among pregnant and breastfed Jordanian women should be clearly stated.

The current situation of other developed and developing countries should also be added.

Explaining why this topic was chosen for analysis in this article is not well written. The benefits of conducting the study to the community should be explained.

The aim should be clearly stated.

4) METHODS:

The characteristics of the study participants should be mentioned as inclusion criteria and exclusion criteria (if any).

Definitions as pregnant woman, women planning to be pregnant, breastfed woman, ……. etc. should be mentioned.

It is advisable to include the questions as per each domain in the methodology. How long did it take to complete the questionnaire(s)? Mention the number of questions in the questionnaire(s). As well as did the participant answer it themselves or with assistance from researcher?

Since you conducted the data collection with online means, what strategy was devised to increase the accuracy of the study and the accuracy of the answers? What types of technique the authors used to keep/control the data quality?

5) RESULTS:

Titles of the tables has to be improved to be more specific.

The correlation between perceptions of the seriousness of COVID-19, perceived benefits, motivation and causes of action of being vaccinated, and hesitancy should be added to the comments of table 4.

It is advisable to add multiple regression analysis at the end of the results.

6) DISCUSSION:

Please start the discussion with the study objective, a short summary of the study, and the main findings.

Discuss by using the scientific reasoning the COVID-19 vaccine hesitancy among pregnant and breastfed Jordanian women in other developing and developed countries with similar context. The manuscript could be greatly strengthened if the authors could compare the findings of the study with other findings and state the reasons for the strengths and weaknesses in each section.

7) CONCLUSION:

It should be specific and based on the findings of the study.

8) STRENGTHS AND LIMITATIONS:

Please analyze the strengths of the study.

Rewrite the limitations of the study.

Reviewers' comments:

Reviewer's Responses to Questions

**Comments to the Author**

1. If the authors have adequately addressed your comments raised in a previous round of review and you feel that this manuscript is now acceptable for publication, you may indicate that here to bypass the “Comments to the Author” section, enter your conflict of interest statement in the “Confidential to Editor” section, and submit your "Accept" recommendation.

Reviewer #1: All comments have been addressed

Reviewer #3: (No Response)

Reviewer #4: (No Response)

Reviewer #5: All comments have been addressed

2. Is the manuscript technically sound, and do the data support the conclusions?

Reviewer #1: Yes

Reviewer #3: Partly

Reviewer #4: Yes

Reviewer #5: Yes

3. Has the statistical analysis been performed appropriately and rigorously? 

Reviewer #1: Yes

Reviewer #3: Yes

Reviewer #4: No

Reviewer #5: Yes

4. Have the authors made all data underlying the findings in their manuscript fully available?

Reviewer #1: Yes

Reviewer #3: Yes

Reviewer #4: Yes

Reviewer #5: Yes

5. Is the manuscript presented in an intelligible fashion and written in standard English?

Reviewer #1: Yes

Reviewer #3: Yes

Reviewer #4: Yes

Reviewer #5: Yes

6. Review Comments to the Author

Reviewer #1: the manuscript is more coherent in terms of methodology and correlates using rigorous tests and comparisons

The results are clearly displayed to show the outcomes in the 2 different categories of women

The implication for for policy and healthcare services are clear and well highlighted which i believe is important in this part of the world given the low covid-19 vaccine uptake among not only women but also health care providers

Reviewer #3: The authors should look into the typo errors before submitting the revision as well as for english. Also the response to comments raise by reviewers are not complete. For the comment regarding data collection the authors didn't add justification in the main text why they don't think its a limited sited of recruitment. Also it is not clear if all the maternity clinic were covered or not. The sampling technique and further increasing the sample may add bias.

Reviewer #4: The article produces important evidence regarding uptake of COVID-19 among women planning to become pregnant, who are pregnant, and who are breastfeeding which is key in formulating polices and recommendations to control the pandemic. However, the article cannot published in its current format due to a number of issues.

Introduction

1. The introduction seems to be long (2 pages). The authors should consider making short and concise ( introduce the topic, what is already known, what is unknown/justification for the study and finally aim/objective)

Methods

Study design

2. The authors mentions that it was “correlational” study design. Why do they refer the study that way? In correlational studies, the study unit is not people but geographical.

3. Some information is missing for sample size calculation. Consult a Statistician for the sample size calculation for cross-sectional studies with a comparison group. Some assumptions especially the two proportions (p1 and p2 ) and there references are missing.

3. “…..women who are planning to be pregnant, pregnant women, and breastfeeding women, and to compare them with other women”. Identifying pregnant and breastfeeding women. What was the distinction between women planning to be pregnant and other women? How were women planning to be pregnancy identified? There is a danger of misclassification.

4. Please include the name of ethical institution and the IRB number

5. How was the barcode/ link for the survey distributed to the participants? Was the scanning done via a mobile phone, bar code scanner etc? Please specify these details sequentially. Was measures were put in place that each participate answers the questionnaire once. What measures were also put to make sure those outside the study area did not participate? With social media link can be easily shared.

Results

6. The sample size calculation assumptions were to compare the outcomes of women who are planning to be pregnant, pregnant women, and breastfeeding women with the “other women”. The “other women” are not coming out in the results showing the comparison

7. The calculated sample size was 847 but you interviewed more than double 1, 874. Any implications and if yes these should be discussed.

8. In Table 1, were age and years of education normally distributed? The mean and SD is reported for normally distributed data, otherwise report the median.

9. Results section 3.2, “In this study, perceptions of COVID-19 seriousness, vaccine hesitancy, perceived benefits, and motivation and causes of action to take the vaccine were all considered as dependent variables whereas other variables were grouping variables”. This should be moved to the methods sections on variables and expanded further. Consider using the term outcome(s) and independent (exposure, explanatory) variables which easier to follow.

Discussion

10. The first paragraph of the discussion should summarise the main findings of the study and later paragraphs should discuss finding by finding.

11. Include strengths of the study

12. What is listed as limitations are not limitations. Usually limitations should include methodological issues which could have affected the validity of the results. Please revise

Others

13. The authors should maintain the sequence “Women planning to become pregnant, who are pregnant, and who are breastfeeding” throughout the paper. In some instances it not followed.

Reviewer #5: The subject matter of this research is very important for public health practice especially maternal and infant health. The objective of the study is clear and a good introduction to the study is provided. I think the discussion section of the study was well presented with the relating of the findings of the study to similar studies conducted in different countries and among different population groups. The researchers did well by comparing the consistent and inconsistent results from other studies. Also, increasing the sample size to a much bigger number makes the results more representative of the population of women in Jordan and therefore, generalization of the results will be more meaningful.

There are, however, a few recommendations that I think will make the manuscript clearer and more meaningful to the reading public. These include the following.

1. Study design: The researchers only indicated that an online cross-sectional correlational study was conducted without enough information as what that study design entails and why that type of study was chosen over study designs. It will beneficial to elaborate more on why you chose that design and what constitute cross-sectional correlational study. Related to the above statement is the lack of description of what G-power software is for the calculation of sample size. Please, provide brief explanation of the software.

2. Ethical consideration: Please, include the ethical approval number and the name of the institution that granted the approval instead of just stating that the institution of the lead researcher. This is section 2.5.

3. Dependent and independent variables: There appears to be an issue of clarity in terms of which variables are dependent and which are independent. You have lumped them up in my opinion. The dependent variable I see here is COVID-19 vaccine hesitancy which in other words not getting vaccinated. The other variables are independent variables in my opinion which are supposed to impact the uptake or otherwise of the dependent variable hesitancy. Please, clarify.

4. I think including history of chronic diseases in the results section detracts from the main objective of the study and has the potential to the focus of the readers from the most important variables in the study, that is the key concepts of health believe model and its impact on covid vaccine uptake among the women in this study.

5. A very important recommendation I have for you is to get professional editing of the manuscript done. Though not the vaccine of the study or my review, there are inconsistencies in the tenses of the sentences as some appear to present terms and others in the past tense. Since the study is completed and this manuscript is a report of the results, it will be appropriate to ensure that everything is written in past tense unless it is a description of something.

Overall, this study is very important and the results will provide baseline data and information appropriate health education program planning and future research.

7. PLOS authors have the option to publish the peer review history of their article (what does this mean?). If published, this will include your full peer review and any attached files.

Reviewer #1: No

Reviewer #3: **Yes: **Mainak Bardhan

Reviewer #4: **Yes: **Richard Makurumidze

Reviewer #5: No

---

## [Author Response · Author response to Decision Letter 1]

8 Feb 2023

Response to Editor and reviewers was submitted in a separate file

---

## [Decision Letter · Decision Letter 2]

15 Mar 2023

PONE-D-22-08549R2COVID-19 Vaccine Hesitancy Among Women Planning for Pregnancy, Pregnant or Breastfeeding in JordanPLOS ONE

Dear Dr. Masa'Deh,

Thank you for submitting your manuscript to PLOS ONE. After careful consideration, we feel that it has merit but does not fully meet PLOS ONE’s publication criteria as it currently stands. Therefore, we invite you to submit a revised version of the manuscript that addresses the points raised during the review process.

We look forward to receiving your revised manuscript.

Kind regards,

Ghada Abdrabo Abdellatif Elshaarawy, M.D

Academic Editor

PLOS ONE

Journal Requirements:

Additional Editor Comments:

Titles of the tables has to be improved to be more specific.

Make sure you thoroughly address the reviewers’ comments and concerns and ensure the manuscript is free of any editorial or grammatical errors.

Reviewers' comments:

Reviewer's Responses to Questions

**Comments to the Author**

1. If the authors have adequately addressed your comments raised in a previous round of review and you feel that this manuscript is now acceptable for publication, you may indicate that here to bypass the “Comments to the Author” section, enter your conflict of interest statement in the “Confidential to Editor” section, and submit your "Accept" recommendation.

Reviewer #5: All comments have been addressed

Reviewer #6: All comments have been addressed

Reviewer #7: (No Response)

Reviewer #8: (No Response)

2. Is the manuscript technically sound, and do the data support the conclusions?

Reviewer #5: Yes

Reviewer #6: Yes

Reviewer #7: No

Reviewer #8: (No Response)

3. Has the statistical analysis been performed appropriately and rigorously? 

Reviewer #5: Yes

Reviewer #6: Yes

Reviewer #7: No

Reviewer #8: (No Response)

4. Have the authors made all data underlying the findings in their manuscript fully available?

Reviewer #5: Yes

Reviewer #6: Yes

Reviewer #7: Yes

Reviewer #8: (No Response)

5. Is the manuscript presented in an intelligible fashion and written in standard English?

Reviewer #5: Yes

Reviewer #6: Yes

Reviewer #7: No

Reviewer #8: No

6. Review Comments to the Author

Reviewer #5: Once again, I have had the opportunity to review the revised version of the manuscript and have seen that the revisions made further improved the manuscript from the introduction to the conclusions especially the methodology, analysis, and discussion. It is an important study not just for COVID-19 vaccine acceptance but many other vaccines and therapeutics for maternal and child health improvement.

Reviewer #6: The paper is revised and addressed all the authors concerns, the paper now is ready for publication.

Reviewer #7: PONE-D-22-08549R2

COVID-19 Vaccine Hesitancy Among Women Planning for Pregnancy, Pregnant or Breastfeeding in Jordan

Comments:

Abstract: It had 419 wording which was too much as the PLOS guidelines. Nothing mentioned in abstract about the questionnaire used in this study. P-values for statistically significant should reported. Abstract should be revised, and only relevant important findings should be reported.

Background:

Pleas change this heading to Introduction.

The statement “The Centers for Disease Control and Prevention (CDC) reported that all individuals at any age can be infected with the virus and their infection can be severe; however, individuals with underlying medical conditions are at higher risk of having more severe consequences of COVID-19, [2, 5].” Reference 2 was not published by CDC.

The authors stated that “ Several COVID-19 vaccines (hereinafter “vaccines”) were developed in an attempt to end the pandemic, four of which received emergency use authorization approval to be used in Jordan, namely: Pfizer/ BioNTech, Sputnik V, Oxford/ AstraZeneca, and Sinopharm [2, 11].” Both references 2 and 11 were about frequently asked questions on COVID-19, nothing about vaccines.

The statement “…availability and accessibility of the vaccine (Limaye et al., 2022)” and “..breastfeeding women (Bianchi et al., 2022)”. For both number of the reference should be used rather than the name of the authors.

The authors stated that “This hesitancy was explained by two main reasons.” Which contradict with their statement at the end of the page “Several factors can affect vaccine hesitancy”, please revise.

The authors stated that “To our knowledge, vaccine hesitancy among women planning for pregnancy, pregnant and breastfeeding was not studied before in Jordan.” This was an assumption as authors were unaware of unpublished work in Jordan in this regard.

Therefore, the introduction (background) of this manuscript should be revised.

Methodology: This section requires major revision.

The questionnaire used in this study had major issues.

The authors stated that “This study implements an online cross-sectional study.” Since the data collection occurred during the COVID-19 pandemic , was it a virtual survey and where was it (home or clinic”.

What were the exclusion criteria? How many women were excluded?

The reasons to increase sample size were not convincing.

Under demographic characteristics sheet “what were the definitions of comfortable, varies and tight of financial status?

The authors stated that “The Arabic Version of Perceptions, Hesitancy, Perceived Benefits, Motivation and Cause of Action About COVID-19 Vaccination Questionnaire This questionnaire has been previously used in studies around the globe (Lin et al., 2020), and has been reported to be a valid and a reliable questionnaire to be used in Arabic communities [16, 21].” Both references did not have the Arabic questionnaire .

Reference 16. Samannodi, M., COVID-19 Vaccine Acceptability Among Women Who are Pregnant or Planning for Pregnancy in Saudi Arabia: A Cross-Sectional Study. Patient Preference and Adherence, 2021. Volume 15: p. 2609-2618. While 21. Lin, Y., et al., Understanding COVID-19 vaccine demand and hesitancy: A nationwide online survey in China. 2020. 14(12): p. e0008961.

Indeed reference 16 used questionnaire published by Almaghaslah D, Alsayari A, Kandasamy G, et al. COVID-19 vaccine hesitancy among young adults in Saudi Arabia: a cross-sectional web-based study. Vaccines. 2021;9(4):330. doi: 10.3390/vaccines9040330.

Almaghaslah D, et al developed the questionnaire on previous systematic review on work published by Lin C, Tu P, Beitsch L. Confidence and receptivity for COVID-19 vaccines: a rapid systematic review. Vaccines 2020;9(1):16. doi: 10.3390/vaccines9010016.

Therefore, the questionnaire used this study should be based on Almaghaslah D, et al work.

The validity of the questionnaires used in this study, and their Cron.bach’s alpha and reliability scores should be reported. The questionnaire should be explained in detail. How many sections in the questionnaire ( Reference 16 had 5 sections while authors reported 4 domain) ? the possible scores for each section / domain must be calculated?

The authors stated “Regarding the second section (participants’ perceptions about COVID-19), the higher the score the weaker the perception about the seriousness of the COVID-19 situation. Regarding the third section (participants’ hesitancy about receiving the COVID-19 vaccine), the higher the score the greater the hesitancy to receive the vaccine. Regarding the fourth section (participants’ perceived benefits of receiving the COVID-19 vaccine), the higher the score the stronger the perception about the perceived benefits of the vaccine. Regarding the last section (participants’ motivations and causes of action for receiving the COVID-19 vaccine), the higher the score the lower the motivation or likelihood to take action to receive the vaccine”. This contradicts with the findings reported in reference 16.

What was/were the health models among the participants’ perception about COVID-19 and vaccination in this study?

What was the sampling technique? How were the data extracted?

Results:

The whole results should be revised once the comments in methodology were incorporated.

Tables: Table 1 was very busy and did not fit in one page. I suggest either to split it or delete unimportant variables in this table. Under Table 1, What was the definition of other women? Under Table 1, the total married women was 492 and this contradict with the total women (428) who were pregnant , planning for pregnancy and breastfeeding. Please revise.

Table 2: It was very busy and did not fit in one page. I suggest either to split it or delete unimportant variables in this table. The data for “However, the results showed no significant difference in perception, hesitancy, perceived benefits, and motivation and causes of action of women according to their residency and financial status” were not shown in Table 2.

I suggest to delete Table 2 and replace it with another Table to calculate the total mean score for each domain of pregnant , planning for pregnancy and breastfeeding and compare it to the total mean score of other women.

Table 3: : It was very busy and did not fit in one page. I suggest either to split it or delete unimportant variables in this table. Furthermore, it was inappropriate statistical analysis.

Authors must do statistical analysis for the mean vaccination perception, hesitancy, perceived benefit and cause of action score stratified by demographic characteristics for women Who are pregnant or planning to be or breastfeeding in order to adjust for all possible confounding factors.

Limitations of the study: It should be explained in details and logical ways. Limitations section should be focused on biases, such as recruitment, personal, recall, and misinterpretation biases as these were possibilities in this study.

General comments: The manuscript was full of typo and in dire need to be edited by someone more proficient in English.

It is crucial to check this manuscript for similarity.

Reviewer #8: Dear authors,

Thank you for sharing this manuscript.

Your study addresses a research question with high relevance. However, the study has been poorly reported. There are many questions and concerns about the study design and implementation. The study would greatly benefit from a professional English language editing.

Please find my comments in the enclosed file.

Good luck with revising the manuscript.

Kind regards,

Saleh Aljadeeah

7. PLOS authors have the option to publish the peer review history of their article (what does this mean?). If published, this will include your full peer review and any attached files.

Reviewer #5: No

Reviewer #6: No

Reviewer #7: No

Reviewer #8: No

---

## [Author Response · Author response to Decision Letter 2]

8 Apr 2023

Responses are attached in a separate file

---

## [Decision Letter · Decision Letter 3]

2 May 2023

PONE-D-22-08549R3COVID-19 Vaccine Hesitancy Among Women Planning for Pregnancy, Pregnant or Breastfeeding in Jordan: A Cross-sectional StudyPLOS ONE

Dear Dr. Masa'Deh,

Thank you for submitting your manuscript to PLOS ONE. After careful consideration, we feel that it has merit but does not fully meet PLOS ONE’s publication criteria as it currently stands. Therefore, we invite you to submit a revised version of the manuscript that addresses the points raised during the review process.

We look forward to receiving your revised manuscript.

Kind regards,

Ghada Abdrabo Abdellatif Elshaarawy, M.D

Academic Editor

PLOS ONE

Journal Requirements:

Additional Editor Comments:

Titles of the tables has to be improved to be more specific.

Reviewers' comments:

Reviewer's Responses to Questions

**Comments to the Author**

1. If the authors have adequately addressed your comments raised in a previous round of review and you feel that this manuscript is now acceptable for publication, you may indicate that here to bypass the “Comments to the Author” section, enter your conflict of interest statement in the “Confidential to Editor” section, and submit your "Accept" recommendation.

Reviewer #5: All comments have been addressed

Reviewer #8: All comments have been addressed

2. Is the manuscript technically sound, and do the data support the conclusions?

Reviewer #5: (No Response)

Reviewer #8: Yes

3. Has the statistical analysis been performed appropriately and rigorously? 

Reviewer #5: Yes

Reviewer #8: Yes

4. Have the authors made all data underlying the findings in their manuscript fully available?

Reviewer #5: Yes

Reviewer #8: (No Response)

5. Is the manuscript presented in an intelligible fashion and written in standard English?

Reviewer #5: Yes

Reviewer #8: Yes

6. Review Comments to the Author

Reviewer #5: After reading reading this manuscript many times and reviewing it for three times, I maintain my review conclusion that the authors have done remarkable job of working hard to improve the entirety of the manuscript from my observation. The methodology, statistical analysis, and discussions are very well developed and presented. Again, the subject matter of this manuscript is important for maternal and child health as well as other important diseases/conditions of public health importance that require vaccine interventions.

Reviewer #8: Dear authors,

Thank you for addressing my comments in your revised manuscript. I recommend publishing your manuscript.

Best wishes

7. PLOS authors have the option to publish the peer review history of their article (what does this mean?). If published, this will include your full peer review and any attached files.

Reviewer #5: No

Reviewer #8: No

---

## [Author Response · Author response to Decision Letter 3]

3 May 2023

Responses are provided in a separate file

---

## [Editor Report · Decision Letter 4]

15 May 2023

COVID-19 Vaccine Hesitancy Among Women Planning for Pregnancy, Pregnant or Breastfeeding Mothers in Jordan: A Cross-sectional Study

PONE-D-22-08549R4

Dear Dr. Masa'Deh,

We’re pleased to inform you that your manuscript has been judged scientifically suitable for publication and will be formally accepted for publication once it meets all outstanding technical requirements.

Kind regards,

Ghada Abdrabo Abdellatif Elshaarawy, M.D

Academic Editor

PLOS ONE
---

## [Editor Report · Acceptance letter]

22 May 2023

PONE-D-22-08549R4 

COVID-19 Vaccine Hesitancy Among Women Planning for Pregnancy, Pregnant or Breastfeeding Mothers in Jordan: A Cross-sectional Study 

Dear Dr. MASA'DEH:

I'm pleased to inform you that your manuscript has been deemed suitable for publication in PLOS ONE. Congratulations! Your manuscript is now with our production department. 

Kind regards, 

on behalf of

Dr. Ghada Abdrabo Abdellatif Elshaarawy 

Academic Editor

PLOS ONE